# Determination of Minimum Doses of Imazamox for Controlling *Xanthium strumarium* L. and *Chenopodium album* L. in Bean (*Phaseolus vulgaris* L.)

**Ramazan Gürbüz [1],*** and **Ömer Yentürk [2]**

[1] Department of Herbology, Faculty of Agriculture, Iğdır University, Iğdır 76000, Turkey
[2] Department of Agricultural Science, Postgraduate Education Institute, Iğdır University, Iğdır 76000, Turkey; omerynt021@gmail.com
* Correspondence: r_grbz@yahoo.com

**Abstract:** This study was conducted to investigate the minimum doses of the imazamox active ingredient (ai) that provide satisfactory efficacy (>90%) against fat hen (*Chenopodium album* L.) and common cocklebur (*Xanthium strumarium* L.). These two weeds are among the most troublesome weeds of bean fields. The minimum dose studies were carried out separately in the 2–4 and 6–8 true leaf stages of both weeds. The experiments were carried out in pots under greenhouse conditions. The experiments were repeated three times. In the first two experiments, the recommended dose of imazamox (100%) together with 75%, 50% and 25% doses were applied to the weeds during the above-mentioned leaf stages. Some pots were left untreated for control. In the third experiments, 12.50% and 6.25% of the recommended doses were also tested. Plant height and the number of leaves were recorded on the 1st, 3rd, 5th, 7th, 14th, 21st and 28th days following the herbicide application. As a result of the studies, it was determined that nearly half the recommended dose (48.18 g ai/da) provides 90% success in the control of common cocklebur (*X. strumarium*) when applied at the 2–4 true leaf stages, while a lower dose (36.11 g ai/da) is required for obtaining the same control when applied at the 6–8 true leaf stages. For the fat hen (*C. album*), only a 17.69 g ai/da application dose was found to provide 90% control at the period of 2–4 true leaves, while 21.21 g ai/da was noted to provide 90% control when applied at the 6–8 true leaf stage. The results suggest that the increase in leaf area reduces the imazamox requirement for the control of *X. strumarium*.

**Keywords:** herbicides; $ED_{90}$; biomass control; dose–response; leaf numbers; plant height

## 1. Introduction

Green bean (*Phaseolus vulgaris* L.) is among the most important crops in the fresh vegetable market. In 2019, the total global area that was devoted for green bean production was about 1.65 M ha, where the total production was estimated to be about 27 M tonnes.. As for many crop species, China is in first place for the production of green bean with a total of 21.7 M tonnes, followed by Indonesia, India and Turkey with a total production of 948 k, 726 k and 596 k tonnes, respectively [1]. Its high content of proteins, flavonoids, vitamin A, dietary fibre and potassium makes the green bean an important nutritional crop for humans [2]. One of the most important problems in green bean production are weeds, which compete with crops for space, water, nutrients and light. Weed control in green beans preferably utilizes post-emergence herbicides rather than pre-emergence [3]. The excessive use of herbicides has been reported to have significant negative impacts on the environment, including water and soil quality, and biodiversity [4–6]. Furthermore, determination of the minimum doses for optimum control [7,8] and critical periods [9] is very important for reducing herbicide use, as part of integrated weed management systems. The use of herbicides is difficult because of the high sensitivity of legumes to most products and the selectivity of herbicides to certain groups of weeds. Minimum

doses are recommended when crops are in good condition and are also associated with a sensitive phase (period) of weed development [10,11]. In a previous study, Stagnari and Pisante [3] recommended that 11 to 28 days after emergence (DAE) is a critical period for obtaining the highest yield in green bean production, which nearly equals to the 2–4 and 6–8 true leaf stages of the weeds. However, it is well known that the growing conditions and planting pattern [12,13]; weed coverage and composition [14,15]; light intensity [16]; and soil characteristics [17] significantly affect the critical period and the required minimum doses of herbicides.

Numerous studies have suggested that the recommended rates of herbicides are set to provide adequate control under various environmental and growing conditions [8,18,19]. Therefore, doses that are lower than the recommended rates could provide satisfactory control under different conditions [8,20–24]. Global total pesticide use was estimated to be 4.1 M tonnes in 2018, with herbicides amounting to 29.5% at 1.2 M tonnes [1]. The two most important weeds causing significant reduction in the green bean yield in the Mediterranean climates of Turkey and Egypt are the fat hen (*Chenopodium album* L.) and common cocklebur (*Xanthium strumarium* L.) [25–27]. *C. album* grows well in temperate regions but becomes a problem in nearly all winter-sown crops that are grown in tropical and subtropical regions [28]. *X. strumarium* is an important weed prevailing in various ecosystems around the world and causing significant losses in crop yield globally [29]. One of the most important herbicide active ingredients (ai) is imazamox (40 g/L), which was recommended for post-emergence application in green beans with a 100 mL/da dose [30–32].

Imazamox is a member of the family imidazolinone which inhibit the acetolactate synthase (ALS) enzyme. It can be absorbed by roots and leaves and easily moves within the plant to the growing points and inhibits ALS activity. This results in blocking the production of branched-chain amino acids and reducing the protein synthesis and cell division in the plants, causing weeds to die. Imazamox is reported to provide effective control against broad leaf weeds [32,33]; however, it works relatively slowly as compared with some other active ingredients. The first symptoms may appear 2 days after application, but the death of the plants may take 2–3 weeks [33]. Imazamox has high persistence in soil and high solubility in water, which gives it high potential to injure the succeeding crops [34]. In a previous study, Soltani et al. [35] reported that the recommended doses of imazethapyr in dry bean can be reduced by combining them with trifluralin. However, no studies are available about the impacts of imazamox specifically on *C. album* and *X. strumarium*. Therefore, the present study aimed to identify the minimum doses of the imazamox active ingredient (ai) that provide satisfactory efficacy (>90%) against *C. album* and *X. strumarium*, which are among the most problematic weeds causing yield loss in green bean production in the Mediterranean Region. The minimum dose studies on these two weeds focused on two different true leaf stages, namely the 2–4 and 6–8 true leaf stages.

## 2. Materials and Methods

### 2.1. Materials

In this study, herbicide, with the active ingredient imazamox—which can be applied before and after emergence—was used. The trade name of this herbicide is Intervix Pro (40 gr/l imazamox SL, BASF®). The recommended post-emergence dose for imazamox in bean fields is 100 mL/da. The dose is a water-soluble concentrate in "SL" formulation. Two weeds were used in the present study: *C. album* and *X. strumarium*, belonging to the families of Chenopodiaceae and Asteraceae, respectively. The seeds of the given weeds were collected from the weed species that were grown from wetlands and roadsides in the provinces of Diyarbakır in the summer of 2020, separated from the dried plant seeds and stored in a room at 10 °C under dark conditions.

The soil from the current research was collected from a cereal cultivated land where no herbicide had been applied in the last three years. The soil organic matter was 3.7%

where the soil texture was clay loam. The pH of the soil that was used in the trial pots was 7.8, the soil lime content was 12.2% and the soil salinity was 0.02%.

## 2.2. Experimental Design and Data Collection

This study was conducted in plastic pots with a diameter of 150 mm, a depth of 13 cm and a volume of 1.600 $cm^3$. Studies were conducted at the two different growing periods (2–4 and 6–8 true leaf stages) of both weeds and all studies were repeated three times. In the first two studies, five different doses of the imazamox herbicide were used. These doses are: 100%, 75%, 50%, 25% and 0% of the recommended dose (100 mL/da) of the herbicide. In the third experiment, 12.50% and 6.25% of the recommended dose of the herbicide were also included in the experimental studies. Each dose (treatment) was used with five replications (pots). A total of six seeds were planted in each pot (filled with the above-mentioned soil), and at the mentioned growing states (2–4 and 6–8 true leaf stages), two healthy weeds of a similar size were left in each pot. The pots were irrigated with a 2-day interval at the beginning of the experiments, and after the growth of the plants, irrigation was performed in the soil based on the needs of the plants.

In the applications of imazamox, the herbicide rate per decare was calculated and applied with a fan-type nozzle (F110-02—yellow flat fan 110° spray nozzle) with a hand sprayer operating at 3 atm pressure. The hand sprayer that was used in the research had a water capacity of 16 L. Since the hand sprayer had a working width of 1 m when working with a single nozzle, the pots were mixed in a 1 m × 4 m (4 $m^2$) area. Nearly 20–40 L of water is recommended for a 1 decare area in herbicide application. Thus, 30 L of water was calibrated in the current study, which suggested that 120 mL of water should be applied on the 4 $m^2$ area where the pots were distributed. The calibration studies suggested that the 120 mL of water fell from the nozzle in 14 s, and for this reason, a speed adjustment was made so that the area where the pots were distributed could be walked in 14 s. After the speed adjustment, the applications were made. Herbicide applications were made from the lowest dose to the highest dose, and the tank of the sprayer was thoroughly cleaned while passing from one dose to the next. After spraying, each pot was labelled and put back into an open field under natural conditions, according to the completely randomized design. Irrigation and light requirements were met according to the climate and environmental conditions.

The day of the imazamox application is day 0. Regular measurements (at 1st, 3rd, 5th, 7th, 14th, 21st and 28th days) were performed for the determination of the plant height and leaf numbers [8]. Moreover, the above-ground plant parts of all weeds (of each pot) were removed at the 28th day and dried at 105 °C for 24 h to determine the dry weight of the samples.

## 2.3. Data Analysis

The effects of reduced doses of imazamox on the plant height and leaf number were determined according to the analysis of variance (ANOVA) test. The mean values of each dose were compared each day. In case of a significant difference, the mean separations were calculated with a Tukey HSD test at $p < 0.05$. These studies were conducted with the SPSS 22.0 software of IBM. Moreover, the dry weight data of the third experiments (with seven different doses) were subjected to the four-parameter log-logistic model, where the model-function relating response Y (dry weight) to the imazamox dose "x" is [36,37]:

$$Y = C + \frac{D - C}{1 + exp\{b \times [\log(x) - \log(ED_{50})]\}}$$

In the above-given model function, C means lower limit, D means upper limit, b represents the slope, and $ED_{50}$ is the dose causing a 50% response. The R environment with the add-on package drc was used for the analysis of the dose–response data [37]. Then, the $ED_{50}$ and b values were used in the pre-defined model ($ED_{90} = ED_{50} \times 9^{1/b}$) to determine the $ED_{90}$, which provides 90% control of the weeds' dry weight.

## 3. Results and Discussions

### 3.1. Effect on Plant Heights

According to the results, it was observed that both the recommended and reduced doses of imazamox had a significant impact on the plant height of *X. strumarium* (Table 1). It was observed that the significant impact of imazamox on the *X. strumarium* plant height first began to appear after 3 days of application, when applied at the 2–4 true leaf stage. The plant height of the *X. strumarium* weed reached from 26 cm to 39 cm in the first experiment and from 25 cm to 38 cm in the second experiment. At the same time, the plant height was around 15–16 cm at 25% of the recommended dose and around 5–6 cm at 100% of the recommended dose. These results clearly show that the plant height can be controlled with the minimum test dose of imazamox. The results of the present research also demonstrate that the impact of imazamox is delayed about 2 days and begins about 5 days after application, when it is applied at the 6–8 true leaf stage (it has an accumulative effect). Again, all doses of imazamox were found to significantly impact the plant height of *X. strumarium*. The height of the control (0% of the recommended dose) plants reached from 35–38 cm to 47–49 cm in 28 days. The plant height slightly increased at the minimum test dose (25%) of imazamox, and the other doses (50%, 75% and 100%) were noted to significantly reduce the plant height. A high consistency was also observed between the first and second trials for both the 2–4 and 6–8 true leaf stage studies. Overall, the results suggested that the reduced doses of imazamox could be used to control the *X. strumarium* weed species. These results are in accordance with the records of Kahramanoğlu and Uygur [8] who suggested similar success for metribuzin on the control of *Amaranthus retroflexus* L. and *Sinapis arvensis* L.

**Table 1.** Impacts of reduced doses of imazamox on the plant height of *Xanthium strumarium* L. applied at two different growing stages.

| Weed Species/Growing Stage | Imazamox Doses | Plant Heights (cm) | | | | | | |
|---|---|---|---|---|---|---|---|---|
| | | Day 1 | Day 3 | Day 5 | Day 7 | Day 14 | Day 21 | Day 28 |
| *X. strumarium*/2–4 true leaf stage (1st experiment) | Control | 26.0 a | 28.0 a | 30.0 a | 31.0 a | 33.0 a | 35.0 a | 39.0 a |
| | 25.00% dose | 20.0 b | 21.0 b | 22.0 bc | 23.0 bc | 18.0 b | 15.0 b | 15.0 b |
| | 50.00% dose | 24.0 ab | 25.0 ab | 26.0 ab | 26.0 ab | 21.0 b | 17.0 b | 15.0 b |
| | 75.00% dose | 19.0 b | 20.0 b | 21.0 bc | 21.0 bc | 20.0 b | 15.0 b | 13.0 b |
| | 100.00% dose | 20.0 b | 20.0 b | 18.0 c | 18.0 c | 15.0 b | 5.0 c | 5.0 c |
| *X. strumarium*/2–4 true leaf stage (2nd experiment) | Control | 25.0 ab | 28.0 a | 29.0 a | 30.0 a | 32.0 a | 34.0 a | 38.0 a |
| | 25.00% dose | 22.0 b | 23.0 ab | 23.0 ab | 25.0 abc | 18.0 b | 14.0 b | 16.0 b |
| | 50.00% dose | 22.0 b | 22.0 b | 24.0 ab | 27.0 ab | 20.0 b | 15.0 b | 15.0 b |
| | 75.00% dose | 22.0 b | 22.0 b | 22.0 b | 22.0 bc | 18.0 b | 12.0 bc | 10.0 bc |
| | 100.00% dose | 28.0 a | 28.0 a | 20.0 b | 19.0 c | 15.0 b | 6.0 c | 6.0 c |
| *X. strumarium*/2–4 true leaf stage (3rd experiment) | Control | 25.2 a | 27.6 a | 29.0 a | 30.2 a | 32.2 a | 34.2 a | 38.0 a |
| | 6.25% dose | 25.8 a | 26.6 b | 26.4 b | 27.4 b | 29.2 b | 31.0 b | 34.0 b |
| | 12.50% dose | 24.8 ab | 24.4 b | 24.0 bc | 23.0 c | 24.2 c | 26.0 c | 28.6 c |
| | 25.00% dose | 21.2 c | 21.6 c | 22.0 c | 23.6 c | 17.8 d | 14.0 d | 15.2 d |
| | 50.00% dose | 22.8 c | 23.2 bc | 24.6 bc | 26.4 b | 20.4 d | 15.8 d | 14.6 d |
| | 75.00% dose | 20.4 c | 20.6 c | 21.2 c | 21.2 cd | 18.6 d | 13.0 d | 11.0 e |
| | 100.00% dose | 23.6 bc | 23.6 bc | 18.8 d | 18.0 d | 14.4 e | 5.4 e | 5.4 f |

**Table 1.** *Cont.*

| Weed Species/Growing Stage | Imazamox Doses | Plant Heights (cm) | | | | | | |
|---|---|---|---|---|---|---|---|---|
| | | Day 1 | Day 3 | Day 5 | Day 7 | Day 14 | Day 21 | Day 28 |
| *X. strumarium*/6–8 true leaf stage (1st experiment) | Control | 38.0 a | 39.0 a | 40.0 a | 41.0 a | 43.0 a | 45.0 a | 49.0 a |
| | 25.00% dose | 31.0 ab | 32.0 b | 32.0 b | 33.0 b | 34.0 b | 34.0 b | 34.0 b |
| | 50.00% dose | 30.0 b | 31.0 b | 32.0 b | 32.0 b | 31.0 b | 28.0 c | 24.0 c |
| | 75.00% dose | 31.0 ab | 32.0 b | 31.0 b | 24.0 c | 18.0 c | 14.0 d | 10.0 d |
| | 100.00% dose | 33.0 ab | 34.0 ab | 30.0 b | 25.0 c | 17.0 c | 12.0 d | 7.0 d |
| *X. strumarium*/6–8 true leaf stage (2nd experiment) | Control | 35.0 a | 36.0 a | 41.0 a | 42.0 a | 43.0 a | 45.0 a | 47.0 a |
| | 25.00% dose | 28.0 a | 28.5 a | 28.5 b | 30.0 b | 32.0 b | 34.0 b | 34.0 b |
| | 50.00% dose | 28.0 a | 29.0 a | 29.0 b | 29.0 b | 28.0 b | 27.0 b | 23.0 c |
| | 75.00% dose | 30.0 a | 31.0 a | 31.0 b | 24.0 b | 17.0 c | 13.0 c | 10.0 d |
| | 100.00% dose | 35.0 a | 36.0 a | 30.0 b | 24.0 b | 16.0 c | 11.0 c | 6.0 d |
| *X. strumarium*/6-8 true leaf stage (3rd experiment) | Control | 37.0 a | 37.8 a | 40.6 a | 41.6 a | 43.0 a | 45.0 a | 48.2 a |
| | 6.25% dose | 36.8 a | 36.2 a | 37.6 b | 38.6 b | 40.0 b | 42.0 ab | 45.0 ab |
| | 12.50% dose | 37.8 a | 37.6 a | 37.8 b | 38.6 b | 39.4 b | 40.8 b | 43.0 b |
| | 25.00% dose | 29.8 b | 30.4 b | 30.4 c | 31.6 c | 33.4 c | 34.4 c | 34.4 c |
| | 50.00% dose | 29.4 b | 30.2 b | 30.8 c | 30.8 c | 29.6 c | 27.6 d | 23.8 d |
| | 75.00% dose | 30.8 b | 31.8 b | 31.4 c | 24.4 d | 17.8 d | 13.6 e | 10.0 e |
| | 100.00% dose | 34.2 ab | 35.2 ab | 30.2 c | 24.6 d | 16.6 d | 11.8 e | 6.8 f |

Different letters next to the mean values represent significant difference at 5% level, according to Tukey's HSD test.

In other words, the results demonstrated that the plant height of the *X. strumarium* weed species began to decrease 5 days after the application of the highest dose and 14 days after the application of the other doses (when applied at the 2–4 true leaf stage). At the same time, the plant height increased in the control treatments (Figure 1). In the first experiments, the plant height of the untreated control weeds was recorded to increase by about 50% in 28 days, with respect to the initial heights. Similar results were noted from the second experiment and the plant height of the control weeds increased by about 52.5% in 28 days. At the same time, the recommended dose (100%) of imazamox decreased the plant height by about 75.8% and 79.1% in the first and second experiments, respectively. The lowest dose of imazamox in the present study (25% dose) was also noted to provide significant influence on the reduction of plant height, which was noted as 24.8% and 27.6% in the first and second experiments, respectively. Imazamox provided better performance at higher doses when applied at the 6–8 true leaf stage. The plant heights of the *X. strumarium* weed species began to decrease 5 days after the application of the two highest doses (75% and 100%), whereas the impact of 50% of the recommended dose appeared after 14 days. The recommended dose of imazamox reduced the plant height by 78.7% and 82.9% in 28 days of application in the first and second experiments, respectively. When applied at the 6–8 true leaf stage, 25% of the recommended dose did not provide good performance over the plant height, but 50% of the recommended dose provided a 19.9% and 17.8% reduction in the plant height in the first and second experiments, respectively.

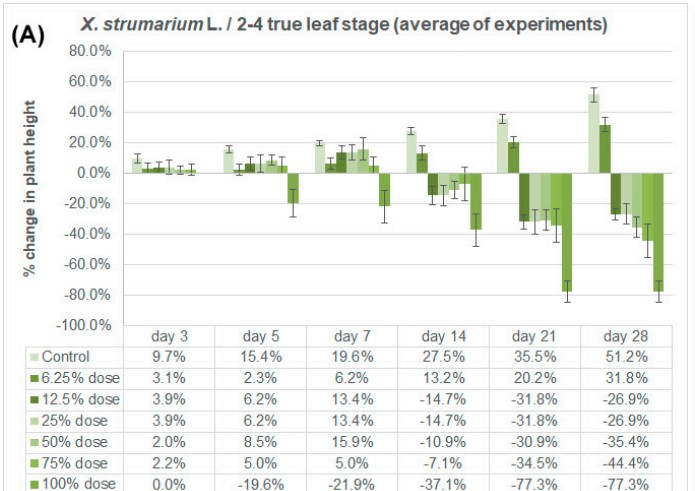
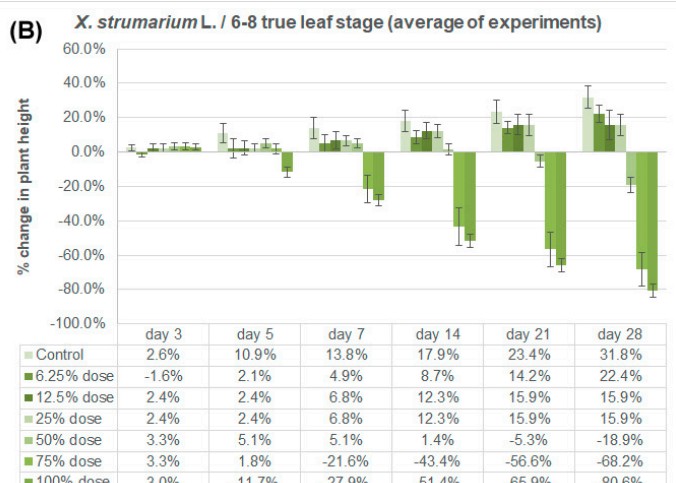

**Figure 1.** Percent (%) change in plant height of *Xanthium strumarium* L. weeds with respect to the initial plant height after treating with imazamox at two different growing stages: (**A**) 2–4 true leaf stages and (**B**) 6–8 true leaf stages.

The results of the third experiment were similar to the results of the first and second trials, while the registered dose of imazamox that was applied in the early period caused a 71.1% reduction in plant height (Table 1; Figure 1A) as compared with the lower doses. While there was no significant decrease in plant height when 6.25% of the recommended dose was applied, a significant shortening of the plant height occurred at higher doses. When applied at a later period (at the 6–8 true leaf stage), no effect was observed at low doses (6.25%, 12%, 25%), while at higher doses (50%, 75%, 100%) this was 19.1% and 100%, respectively. Plant-height shortening occurred at rates of 67.8 and 80.1%.

Studies with *C. album* produced similar results with the *X. strumarium*. It was observed that every dose of imazamox significantly impacted the plant height of *C. album* (Table 2). The significant difference between the control treatment and the herbicide doses was first observed after 5 days of application. Similar findings were observed in both the first and second trials. According to the results, the plant height of the *C. album* weed in the control treatments reached from 2.9–2.9 cm to 11.0–11.1 cm in 28 days for both trials with the 2–4 true leaf stage. In the same trials, the plant height also slightly increased at the lowest dose; the height was noted to reach from 2.4–2.4 to 3.0 cm in 28 days. It is clear from these results that even the 25% dose of imazamox can significantly impact the *C. album* plant height and reduce its growth.

**Table 2.** Impacts of reduced doses of imazamox on the plant height of *Chenopodium album* L. applied at two different growing stages.

| Weed Species/Growing Stage | Imazamox Doses | Plant Heights (cm) | | | | | | |
|---|---|---|---|---|---|---|---|---|
| | | Day 1 | Day 3 | Day 5 | Day 7 | Day 14 | Day 21 | Day 28 |
| *C. album*/2–4 true leaf stage (1st experiment) | Control | 2.9 a | 3.2 a | 3.5 a | 4.0 a | 5.1 a | 7.0 a | 11.1 a |
| | 25.00% dose | 2.4 b | 2.6 bc | 2.8 b | 3.0 b | 3.0 b | 3.0 b | 3.0 b |
| | 50.00% dose | 2.2 b | 2.3 c | 2.4 c | 2.4 c | 2.2 c | 2.2 c | 2.2 c |
| | 75.00% dose | 2.4 b | 2.5 c | 2.5 bc | 2.5 c | 2.0 c | 1.9 c | 1.9 c |
| | 100.00% dose | 2.9 a | 2.9 ab | 2.8 b | 2.7 bc | 2.2 c | 2.2 c | 2.2 c |

**Table 2.** *Cont.*

| Weed Species/Growing Stage | Imazamox Doses | Plant Heights (cm) | | | | | | |
|---|---|---|---|---|---|---|---|---|
| | | Day 1 | Day 3 | Day 5 | Day 7 | Day 14 | Day 21 | Day 28 |
| *C. album*/2–4 true leaf stage (2nd experiment) | Control | 2.9 a | 3.1 a | 3.4 a | 3.8 a | 5.0 a | 7.0 a | 11.1 a |
| | 25.00% dose | 2.4 b | 2.6 bc | 2.8 b | 3.0 b | 3.0 b | 3.0 b | 3.0 b |
| | 50.00% dose | 2.2 b | 2.3 d | 2.4 c | 2.4 c | 2.1 c | 2.1 c | 2.1 c |
| | 75.00% dose | 2.4 b | 2.4 cd | 2.5 c | 2.4 c | 2.0 c | 2.0 c | 2.0 c |
| | 100.00% dose | 2.8 a | 2.9 ab | 2.9 b | 2.8 bc | 2.2 c | 2.2 c | 2.2 c |
| *C. album*/2–4 true leaf stage (3rd experiment) | Control | 2.7 a | 2.9 a | 3.3 a | 3.7 a | 4.9 a | 6.8 a | 10.9 a |
| | 6.25% dose | 2.6 a | 2.7 a | 2.8 b | 3.2 b | 4.1 b | 5.8 b | 9.3 b |
| | 12.50% dose | 2.5 a | 2.5 a | 2.8 b | 3.2 b | 4.1 b | 5.4 b | 4.9 c |
| | 25.00% dose | 2.2 b | 2.4 ab | 2.6 b | 2.8 bc | 2.8 c | 2.8 c | 2.8 d |
| | 50.00% dose | 2.0 b | 2.1 b | 2.2 c | 2.2 c | 1.9 d | 1.9 d | 1.9 e |
| | 75.00% dose | 2.2 b | 2.3 b | 2.3 c | 2.3 c | 1.8 d | 1.7 d | 1.7 e |
| | 100.00% dose | 2.6 a | 2.7 a | 2.7 b | 2.5 c | 2.0 d | 2.0 d | 2.0 e |
| *C. album*/6–8 true leaf stage (1st experiment) | Control | 6.0 a | 6.5 a | 7.5 a | 9.6 a | 16.0 a | 22.0 a | 32.0 a |
| | 25.00% dose | 4.0 b | 5.0 b | 5.1 bc | 4.5 bc | 4.5 b | 4.0 b | 3.0 b |
| | 50.00% dose | 5.7 ab | 5.9 ab | 5.6 b | 5.0 b | 4.0 b | 3.0 b | 2.0 b |
| | 75.00% dose | 5.0 ab | 5.2 ab | 4.8 bc | 4.8 b | 2.0 c | 0.0 c | 0.0 c |
| | 100.00% dose | 5.1 ab | 5.1 ab | 4.0 c | 3.2 c | 0.0 d | 0.0 c | 0.0 c |
| *C. album*/6–8 true leaf stage (2nd experiment) | Control | 8.0 a | 9.6 a | 12.0 a | 16.0 a | 25.6 a | 28.4 a | 35.0 a |
| | 25.00% dose | 7.2 bc | 8.0 b | 8.0 b | 8.0 b | 8.0 b | 8.0 b | 8.0 b |
| | 50.00% dose | 6.8 bc | 6.8 bc | 6.8 b | 6.8 b | 6.8 b | 6.8 b | 6.6 b |
| | 75.00% dose | 6.4 c | 6.4 c | 6.4 b | 6.4 b | 6.4 b | 6.4 b | 6.4 b |
| | 100.00% dose | 8.0 a | 8.0 b | 8.0 b | 8.0 b | 8.0 b | 8.0 b | 8.0 b |
| *C. album*/6–8 true leaf stage (3rd experiment) | Control | 7.2 a | 8.4 a | 9.8 a | 13.0 a | 20.8 | 25.6 a | 33.8 a |
| | 6.25% dose | 7.2 a | 7.4 b | 8.8 b | 11.0 b | 17.8 a | 21.6 b | 28.8 b |
| | 12.50% dose | 7.2 a | 6.8 c | 7.8 c | 7.4 c | 9.4 b | 10.2 c | 10.8 c |
| | 25.00% dose | 5.6 b | 6.8 c | 6.8 d | 6.4 c | 6.4 c | 6.2 d | 5.8 d |
| | 50.00% dose | 6.6 ab | 6.6 c | 6.4 d | 6.0 c | 5.6 c | 5.2 d | 4.4 d |
| | 75.00% dose | 6.2 b | 6.2 d | 6.2 d | 6.2 c | 4.4 c | 3.2 d | 3.2 d |
| | 100.00% dose | 7.0 a | 7.0 c | 6.0 e | 6.0 c | 4.0 c | 4.0 d | 4.0 d |

Different letters next to the mean values represent significant difference at 5% level, according to Tukey's HSD test.

No significant difference was noted among the other three higher doses, but they were found to significantly reduce the plant height, or in other words, prevent plant growth. On the other hand, similar to *X. strumarium*, the impact of imazamox on *C. album* was found to be higher when it was applied later at the 6–8 true leaf stage. The plant height increased from 6–8 to 32–35 cm in control weeds, while the plant height decreased from 5–8 to 0–4 cm in the 100% dose of imazamox 28 days after application (in the first and third trials). No change was observed in the plant height in the second trials. In the first trials, 28 days after application, no significant impact was noted between the two highest doses of imazamox, while the other two doses (50% and 25%) had slightly less impact on the plant height. In the second trials, no significant difference was found among all the doses of imazamox, showing that the reduced dose has a similar impact on the plant height of *C. album*, an impact that is as high as the recommended dose. The current results are in accordance with several previous studies that reported that the reduced doses of different herbicides could provide satisfactory control of the weeds [8,21,23,38–40]. Similar results were also noted by Dogan and Hurle [20], and Auskalnis and Kadzys [22], but they indicated that the impacts of reduced doses of herbicides could diminish with the increase in the development stages

of weeds. However, in the current study, the increase in the development stages could not reduce the impact of imazamox; instead, the impact increased. This is because of the characteristics of the recommended herbicide in that it requires high retention by the weeds. This characteristic is discussed in the final section under the evaluation of dry weight.

The evaluation of the results about the percent change in the plant height of *C. album* also suggested that the three highest doses (except the lowest one) provide a significant reduction in the plant height when applied at both (2–4 and 6–8 true) leaf stages (Figure 2). Similar to *X. strumarium*, when imazamox is applied at the 2–4 true leaf stage, the impact of the highest dose begins after 5 days, while the impacts of the other doses begin to be observed after 14 days. The results of the two separate experiments were also found to have similarities. The percent reductions in the plant height of *C. album* weeds were noted to be 22.6% and 23.9% in the two separate experiments at 100% of the recommended dose. The impact of the imazamox doses on the plant height of *C. album* was different when applied at the 6–8 true leaf stage. In the first treatments, the three highest doses provided good performance in reducing the growth of weeds. The two highest doses were found to provide a 100% reduction in the plant height. However, in the second experiments, the plant height was noted to not change during the 28 days of observations. This is also an important finding, where the untreated control plants showed a 337.5% increase in the plant height.

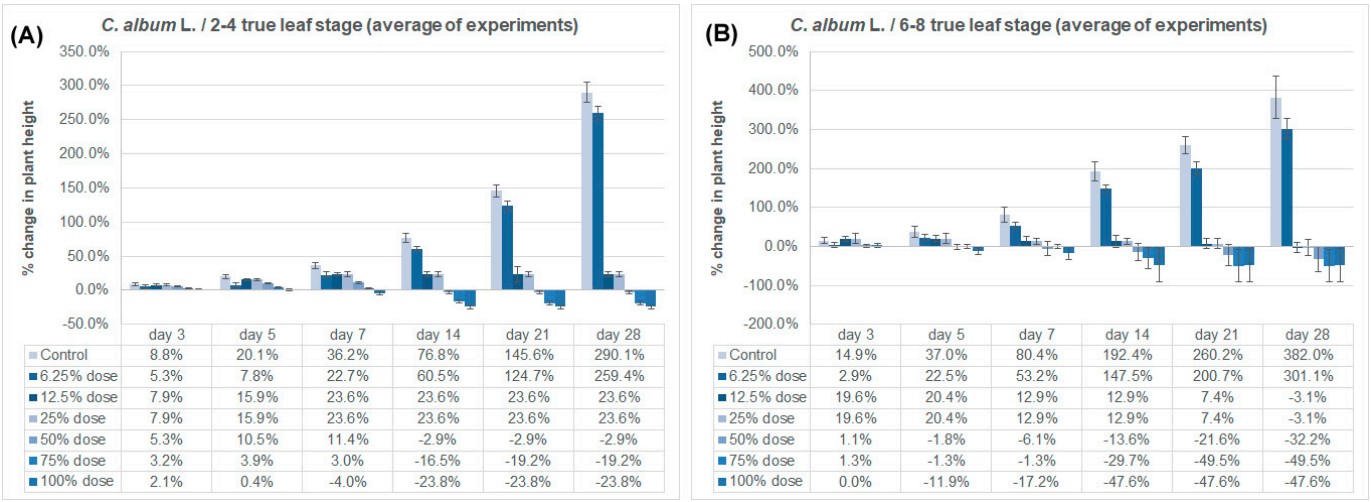

**Figure 2.** Percent (%) change in plant height of *Chenopodium album* L. weeds with respect to the initial plant height after treating with imazamox at two different growing stages: (**A**) 2–4 true leaf stages and (**B**) 6–8 true leaf stages.

### 3.2. Effect on True Leaf Numbers

The impacts of the reduced doses of imazamox on the leaf number were found to be in accordance with the plant height results. When imazamox was applied at the 2–4 true leaf stage, the difference among the treatments was first observed after 3 days in both trials (Table 3). It was noted that the leaf number of the control weeds increased from 4.00 to 14.80 in 28 days, while the number of leaves decreased from 3.60–4.00 to 0.00–4.00 in the herbicide trials. Among the herbicide-applied weeds, only the weeds receiving the lowest dose (25%) were observed to have leaves, while the others dried up and had no leaves. A similar impact on the leaf numbers was observed when imazamox was applied at the later stages (6–8 true leaf stage). However, at that time, the impact of 50% of the herbicide dose was found to be less. Moreover, the impact of the herbicide was somewhat delayed, and the first significant difference was observed 5 days after application, not 3 days. The overall results support the findings of Kahramanoğlu and Uygur [8] who suggested that the reduced doses of metribuzin significantly reduced the leaf number of *A. retroflexus* and *S. arvensis*.

**Table 3.** Impacts of reduced doses of imazamox on the leaf numbers of *Xanthium strumarium* L. applied at two different growing stages.

| Weed Species/Growing Stage | Imazamox Doses | Leaf Numbers | | | | | | |
|---|---|---|---|---|---|---|---|---|
| | | Day 1 | Day 3 | Day 5 | Day 7 | Day 14 | Day 21 | Day 28 |
| *X. strumarium*/2–4 true leaf stage (1st experiment) | Control | 4.0 a | 6.0 a | 7.0 a | 8.0 a | 10.0 a | 12.4 a | 14.8 a |
| | 25.00% dose | 3.6 a | 3.6 b | 4.8 c | 4.8 b | 5.6 b | 3.0 b | 3.0 b |
| | 50.00% dose | 4.0 a | 4.0 b | 6.0 b | 4.8 b | 3.0 c | 0.0 c | 0.0 c |
| | 75.00% dose | 4.0 a | 4.0 b | 4.0 d | 3.0 c | 0.0 d | 0.0 c | 0.0 c |
| | 100.00% dose | 4.0 a | 4.0 b | 2.0 e | 0.0 d | 0.0 d | 0.0 c | 0.0 c |
| *X. strumarium*/2–4 true leaf stage (2nd experiment) | Control | 3.6 a | 5.0 a | 6.8 a | 8.0 a | 10.0 a | 12.0 a | 15.0 a |
| | 25.00% dose | 3.6 a | 3.6 b | 4.6 b | 4.6 b | 5.6 b | 4.0 b | 4.0 b |
| | 50.00% dose | 4.0 a | 4.0 b | 6.0 a | 5.0 b | 2.0 c | 0.0 c | 0.0 c |
| | 75.00% dose | 4.0 a | 4.0 b | 4.4 b | 3.0 c | 0.0 d | 0.0 c | 0.0 c |
| | 100.00% dose | 4.0 a | 4.0 b | 2.4 c | 0.0 d | 0.0 d | 0.0 c | 0.0 c |
| *X. strumarium*/2–4 true leaf stage (3rd experiment) | Control | 4.0 a | 5.8 a | 7.0 a | 8.0 a | 9.8 a | 12.4 a | 15.0 a |
| | 6.25% dose | 4.0 a | 5.2 a | 6.6 b | 7.6 a | 9.4 a | 10.6 b | 12.0 b |
| | 12.50% dose | 4.0 a | 4.6 b | 5.6 c | 6.6 b | 7.2 b | 7.4 c | 7.2 c |
| | 25.00% dose | 3.6 a | 3.8 b | 4.8 c | 4.8 c | 5.4 c | 3.0 d | 3.0 d |
| | 50.00% dose | 4.0 a | 4.2 b | 6.0 b | 4.8 c | 3.2 d | 0.0 e | 0.0 e |
| | 75.00% dose | 4.0 a | 4.2 b | 4.2 cd | 3.2 d | 0.0 e | 0.0 e | 0.0 e |
| | 100.00% dose | 4.0 a | 4.0 b | 2.0 d | 0.0 e | 0.0 e | 0.0 e | 0.0 e |
| *X. strumarium*/6–8 true leaf stage (1st experiment) | Control | 7.0 a | 8.0 a | 10.0 a | 12.0 a | 14.0 a | 15.2 a | 16.4 a |
| | 25.00% dose | 7.0 a | 8.0 a | 9.0 a | 8.0 b | 8.6 b | 8.6 b | 9.0 b |
| | 50.00% dose | 7.0 a | 8.0 a | 7.0 b | 5.0 c | 6.0 c | 6.4 c | 7.2 c |
| | 75.00% dose | 7.0 a | 7.0 ab | 7.0 b | 5.0 c | 2.0 d | 0.0 d | 0.0 d |
| | 100.00% dose | 6.0 a | 6.0 b | 6.0 b | 2.0 d | 0.0 e | 0.0 d | 0.0 d |
| *X. strumarium*/6–8 true leaf stage (2nd experiment) | Control | 6.0 a | 8.0 a | 10.0 a | 10.0 a | 12.0 a | 13.4 a | 15.6 a |
| | 25.00% dose | 6.0 a | 8.0 a | 9.0 a | 8.0 b | 8.4 b | 9.0 b | 10.0 b |
| | 50.00% dose | 6.0 a | 8.0 a | 9.0 a | 8.0 b | 6.4 c | 5.0 c | 10.0 b |
| | 75.00% dose | 6.0 a | 7.0 ab | 7.0 b | 5.0 c | 2.4 d | 0.0 d | 0.0 c |
| | 100.00% dose | 6.0 a | 6.0 b | 6.0 b | 1.6 d | 0.0 e | 0.0 d | 0.0 c |
| *X. strumarium*/6–8 true leaf stage (3rd experiment) | Control | 6.0 a | 7.8 a | 10.0 a | 10.2 a | 12.0 a | 13.4 a | 15.6 a |
| | 6.25% dose | 6.0 a | 7.6 a | 9.2 b | 9.8 a | 11.6 a | 12.0 b | 14.0 a |
| | 12.50% dose | 6.0 a | 7.4 a | 8.4 c | 8.6 b | 9.0 b | 10.0 c | 10.8 b |
| | 25.00% dose | 6.0 a | 8.0 a | 9.0 b | 8.2 b | 8.8 b | 8.8 d | 10.0 b |
| | 50.00% dose | 6.0 a | 8.0 a | 9.0 b | 8.0 b | 6.6 c | 4.8 e | 10.0 b |
| | 75.00% dose | 6.0 a | 6.8 a | 7.0 d | 5.0 c | 2.4 d | 0.0 f | 0.0 c |
| | 100.00% dose | 6.0 a | 6.0 a | 6.2 d | 1.8 d | 0.2 e | 0.0 f | 0.0 c |

Different letters next to the mean values represent significant difference at 5% level, according to Tukey's HSD test.

Similar to the plant height, the leaf numbers of the *X. strumarium* weeds were also noted to be significantly affected by the imazamox doses at both application times (Figure 3). When imazamox was applied at the 2–4 true leaf stage, the three highest doses were found to provide a 100.0% reduction in the number of leaves, whereas the untreated control plants had about a 270.0% and 323.3% increase in the leaf numbers in the first and second experiments, respectively, in 28 days. A change in the application time of imazamox was also noted to change the impact of the herbicide. When imazamox was applied at a later stage (6–8 true leaf stage), the same impact was noted only from the two highest doses, instead of the three highest doses. The *X. strumarium* weeds that were applied with 50% of the recommended dose were observed to enter a sleeping state and recuperate after

21 days. These results are in accordance with the plant height results, explained above, where the plant height began to increase after a period of application.

**Figure 3.** Percent (%) change in leaf numbers of *Xanthium strumarium* L. weeds with respect to the initial leaf numbers after treating with imazamox at two different growing stages: (**A**) 2–4 true leaf stages and (**B**) 6–8 true leaf stages.

The results that were obtained from the third trial which was performed at the 2–4 true leaf stage of *X. strumarium* were similar to the results of the first and second trials. In the third trial, the number of leaves on the weeds increased, as in the other trials, at the 6.25% and 12.5% doses of imazamox that were applied in the early period of *X. strumarium*, different from the other two trials. In all three trials, the highest number of leaves were followed by the control plant in the first two trials, followed by a 25% application dose, and in the third trial, the control continued as 6.25%, 12.5% and 25% doses, respectively.

The results for *C. album* were similar to the results for *X. strumarium*, but a better performance was noted for the herbicide doses (Table 4). The results for leaf number were also found to agree with the results for plant height. When imazamox was applied at the 2–4 true leaf stage, the difference among the treatments was first observed after 5 days in both trials. This was 3 days for the *X. strumarium*. It was noted that the leaf number of the control weeds increased from 3.6 to 17.0–17.2 in 28 days, while the number of leaves decreased from 3.6–4.0 to 0.0–2.6 in the herbicide trials. All doses of the herbicides were noted to reduce the leaf numbers, showing the significant impact on the weeds. A similar impact on the leaf numbers was observed when imazamox was applied at the later stages (6–8 true leaf stage). When the herbicide was applied at later stages, the leaf numbers of the control weeds increased from 7.6 to 65.0 in 28 days. However, the leaf numbers were noted to only be around 9.2–11.0 in the weeds that were treated with the lowest dose of imazamox (25%). The other doses caused a significant reduction in the leaf numbers, while no leaves were counted from the two highest doses of imazamox (100% and 75%).

Percent change in plant heights of *C. album* weeds with respect to the initial plant height after treating with imazamox are given in Figure 4. It is clear from the results that the 75% and 100% of the recommended doses have 100% impact on the leaf numbers of the weeds when applied at the both stages (2–4 and 6–8 true leaf stage). Moreover, the impact of 50% dose was noted to decrease when applied at the later stage (6–8), but it is still valuable. The 50% of the recommended dose was noted to reduce the leaf numbers about 50.0–65.0% when applied at 2–4 true leaf stage; whereas the reduction percentage in the leaf numbers was between 33.3% and 41.1% when applied at the 6–8 true leaf stage. However, comparison with the untreated control weeds, where the leaf numbers increased about 768.6%, showed that the 50% of the recommended dose also have a significant and acceptable influence on the *C. album* weeds.

**Table 4.** Impacts of reduced doses of imazamox on the leaf numbers of *Chenopodium album* L. applied at two different growing stages.

| Weed Species/Growing Stage | Imazamox Doses | Leaf Numbers | | | | | | |
|---|---|---|---|---|---|---|---|---|
| | | Day 1 | Day 3 | Day 5 | Day 7 | Day 14 | Day 21 | Day 28 |
| *C. album*/2–4 true leaf stage (1st experiment) | Control | 3.6 a | 3.6 a | 4.8 b | 6.0 a | 9.2 a | 12.0 a | 17.2 a |
| | 25.00% dose | 3.6 a | 3.6 a | 4.0 b | 4.6 b | 3.6 b | 3.0 b | 2.6 b |
| | 50.00% dose | 4.0 a | 4.0 a | 6.0 a | 5.4 ab | 4.2 b | 3.0 b | 2.0 b |
| | 75.00% dose | 4.0 a | 4.0 a | 4.4 b | 4.4 b | 0.0 c | 0.0 c | 0.0 c |
| | 100.00% dose | 4.0 a | 4.0 a | 4.0 b | 2.0 c | 0.0 c | 0.0 c | 0.0 c |
| *C. album*/2–4 true leaf stage (2nd experiment) | Control | 3.6 a | 3.6 a | 4.8 ab | 6.0 a | 9.0 a | 12.4 a | 17.0 a |
| | 25.00% dose | 4.0 a | 4.0 a | 5.0 a | 5.4 a | 5.0 b | 4.0 b | 2.6 b |
| | 50.00% dose | 3.6 a | 3.6 a | 4.0 b | 4.0 b | 3.0 c | 2.2 c | 1.2 bc |
| | 75.00% dose | 4.0 a | 4.0 a | 4.0 b | 2.6 c | 0.0 d | 0.0 d | 0.0 c |
| | 100.00% dose | 4.0 a | 4.0 a | 4.0 b | 2.0 c | 0.0 d | 0.0 d | 0.0 c |
| *C. album*/2–4 true leaf stage (3rd experiment) | Control | 3.4 b | 3.4 b | 4.6 b | 5.8 a | 9.0 a | 11.8 a | 17.8 a |
| | 6.25% dose | 3.8 a | 4.6 a | 5.6 a | 6.2 a | 8.6 a | 10.6 b | 15.2 b |
| | 12.50% dose | 3.6 a | 3.8 a | 4.4 b | 5.0 b | 4.6 b | 3.4 c | 2.8 c |
| | 25.00% dose | 3.8 a | 3.6 ab | 4.2 b | 4.6 b | 3.6 b | 2.8 c | 2.4 c |
| | 50.00% dose | 3.8 a | 4.0 a | 5.8 a | 5.4 ab | 4.2 b | 2.8 c | 2.0 c |
| | 75.00% dose | 3.8 a | 4.0 a | 4.6 b | 4.2 bc | 0.2 c | 0.0 d | 0.0 d |
| | 100.00% dose | 3.8 a | 4.0 a | 4.0 b | 2.2 c | 0.2 c | 0.0 d | 0.0 d |
| *C. album*/6–8 true leaf stage (1st experiment) | Control | 7.6 a | 9.6 a | 11.8 a | 17.0 a | 28.8 a | 45.2 a | 65.0 a |
| | 25.00% dose | 6.4 ab | 7.8 b | 7.8 b | 9.2 b | 10.2 b | 9.2 b | 9.2 b |
| | 50.00% dose | 6.0 b | 6.5 bc | 7.0 b | 7.0 c | 6.0 c | 5.0 c | 4.0 c |
| | 75.00% dose | 7.2 ab | 7.2 bc | 7.4 b | 7.4 bc | 4.0 cd | 3.0 c | 0.0 d |
| | 100.00% dose | 6.0 b | 6.0 c | 5.0 c | 5.0 d | 2.0 d | 0.0 d | 0.0 d |
| *C. album*/6–8 true leaf stage (2nd experiment) | Control | 7.6 ab | 9.6 a | 11.8 a | 17.0 a | 28.8 a | 45.2 a | 65.0 a |
| | 25.00% dose | 6.8 ab | 7.9 bc | 9.2 b | 10.0 b | 11.1 b | 11.0 b | 11.0 b |
| | 50.00% dose | 6.8 ab | 7.8 bc | 8.0 b | 8.0 c | 6.9 c | 5.4 c | 4.0 c |
| | 75.00% dose | 6.4 b | 6.4 c | 6.0 c | 5.4 d | 4.2 d | 0.0 d | 0.0 d |
| | 100.00% dose | 8.0 a | 8.0 b | 6.0 c | 4.0 d | 0.0 e | 0.0 d | 0.0 d |
| *C. album*/6–8 true leaf stage (3rd experiment) | Control | 7.6 a | 9.6 a | 11.8 a | 17.0 a | 28.8 a | 45.2 a | 65.0 a |
| | 6.25% dose | 7.8 a | 9.4 a | 12.4 a | 16.2 a | 21.2 b | 28.0 b | 35.0 b |
| | 12.50% dose | 7.6 a | 8.6 b | 9.8 b | 11.0 b | 13.8 c | 15.2 c | 17.2 c |
| | 25.00% dose | 7.0 a | 8.0 b | 8.6 b | 8.0 c | 9.8 d | 10.6 d | 11.0 d |
| | 50.00% dose | 7.0 a | 7.8 bc | 8.0 b | 8.0 c | 6.9 e | 5.2 e | 4.1 e |
| | 75.00% dose | 6.6 a | 6.4 c | 6.0 c | 5.2 d | 4.0 f | 0.0 f | 0.0 f |
| | 100.00% dose | 7.8 a | 8.0 b | 6.2 c | 4.2 d | 0.0 g | 0.0 f | 0.0 f |

Different letters next to the mean values represent significant difference at 5% level, according to Tukey's HSD test.

Examination of the all results from the three separate experiments of the *C. album*, at 2–4 true-leaf stages, showed that the results were similar, and the results of the third experiment were the same as the first and second experiments. In the third trial, the number of leaves did not increase after the 7th day, as in the other trials. The highest leaves were obtained from the control treatments in the first two trials, and followed by the 25% of the recommended herbicide dose; moreover, in the third trial, the highest leaves were noted from the control treatment again and was followed by the 6.25%, 12.5% and 25% of the recommended herbicide doses. These results, which were obtained at the stage of 2–4 true leaves of the weed, were also found to be similar with the results of the 6–8 true leaves stage, but it was observed that the increase in the number of leaves stopped after

the 14th day, not at the 7th day at the stage of 6–8 true leaves. It has been observed that the dose to be used should increase as the *C. album* weed develops. When the results of two different plant experiments are compared with each other, it is seen that the resistance of *C. album* to the imazamox active substance is higher than that of *X. strumarium*.

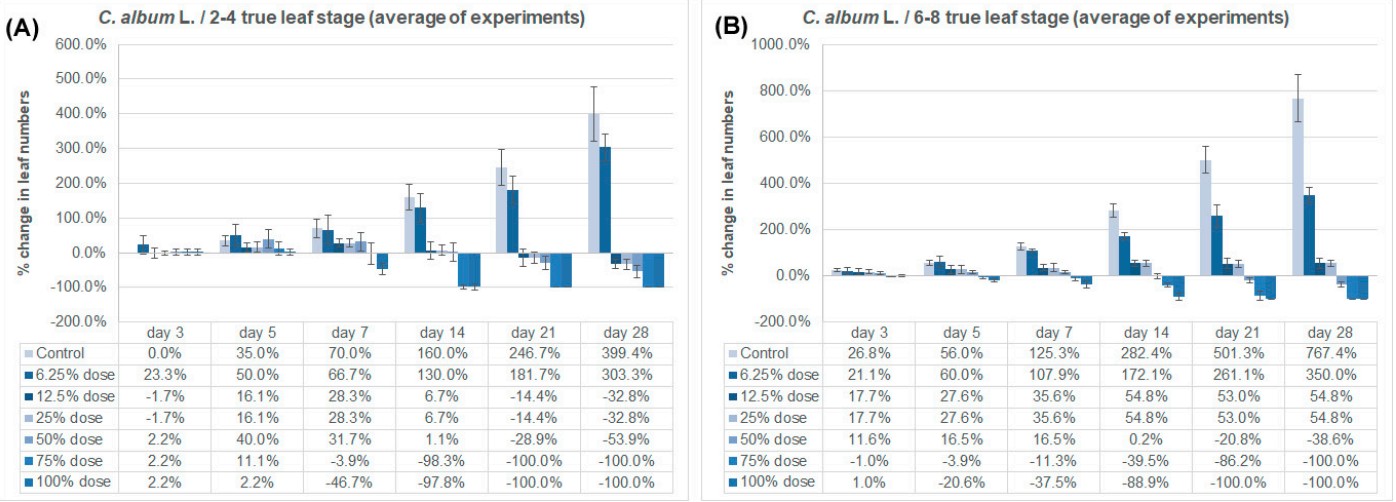

**Figure 4.** Percent (%) change in plant height of *Chenopodium album* L. weeds with respect to the initial plant height after treating with imazamox at two different growing stages: (**A**) 2–4 true leaf stages and (**B**) 6–8 true leaf stages.

### 3.3. Four-Parameter Log-Logistic Model and Minimum Doses

The dose–response (four-parameter log-logistic model) results are presented in Table 5 and the fitted curves for the model are given in Figure 5. The *p*-value of all treatments shows that the tested model is suitable for both weeds at both leaf stages. According to the obtained results, the highest $ED_{50}$ was found for *X. strumarium*/2–4 true leaf stage, while the lowest $ED_{50}$ was found for C. album/6–8 true leaf stage. The dry weights of the third trials were used in the curve fitting and for the estimation of the $ED_{90}$. The results showed that the later leaf stages of *X. strumarium* required lower herbicide dose for reaching a satisfactory control (90% reduction in the dry weight). The results suggest that a 46.18 g ai/da imazamox dose could provide 90% efficacy in the control of *X. strumarium* when applied at the 2–4 true leaf stage, and that the required imazamox dose is lesser when applied at the 6–8 true leaf stage and 36.11 g ai/da is sufficient. Similar results were found for the *C. album*; the required imazamox doses for obtaining a satisfactory control at the 2–4 and 6–8 true leaf stages were noted to be 17.69 and 21.21 g ai/da, respectively.

**Table 5.** Parameters of dose–response curve for imazamox and effective doses (g ai/da) providing 90% control of the weeds together with the lack-of-fit test *p*-values for the model comparison with ANOVA.

| Weed Species | Parameters | | | | $ED_{90}$ | *p*-Value |
|---|---|---|---|---|---|---|
| | C | D | b | $ED_{50}$ | | |
| *X. strumarium*/2–4 true leaf stage | 0.23 | 8.41 | 2.25 | 17.36 | 46.18 | 0.9487 [ns] |
| *X. strumarium*/6–8 true leaf stage | 0.79 | 19.21 | 2.52 | 15.10 | 36.11 | 0.9157 [ns] |
| *C. album*/2–4 true leaf stage | 0.00 | 1.02 | 3.07 | 8.66 | 17.69 | 0.9336 [ns] |
| *C. album*/6–8 true leaf stage | −0.01 | 2.95 | 2.03 | 7.17 | 21.21 | 0.7314 [ns] |

*p*-value of lack-of-fit, if >0.05 (not significant) means that the model fits well.

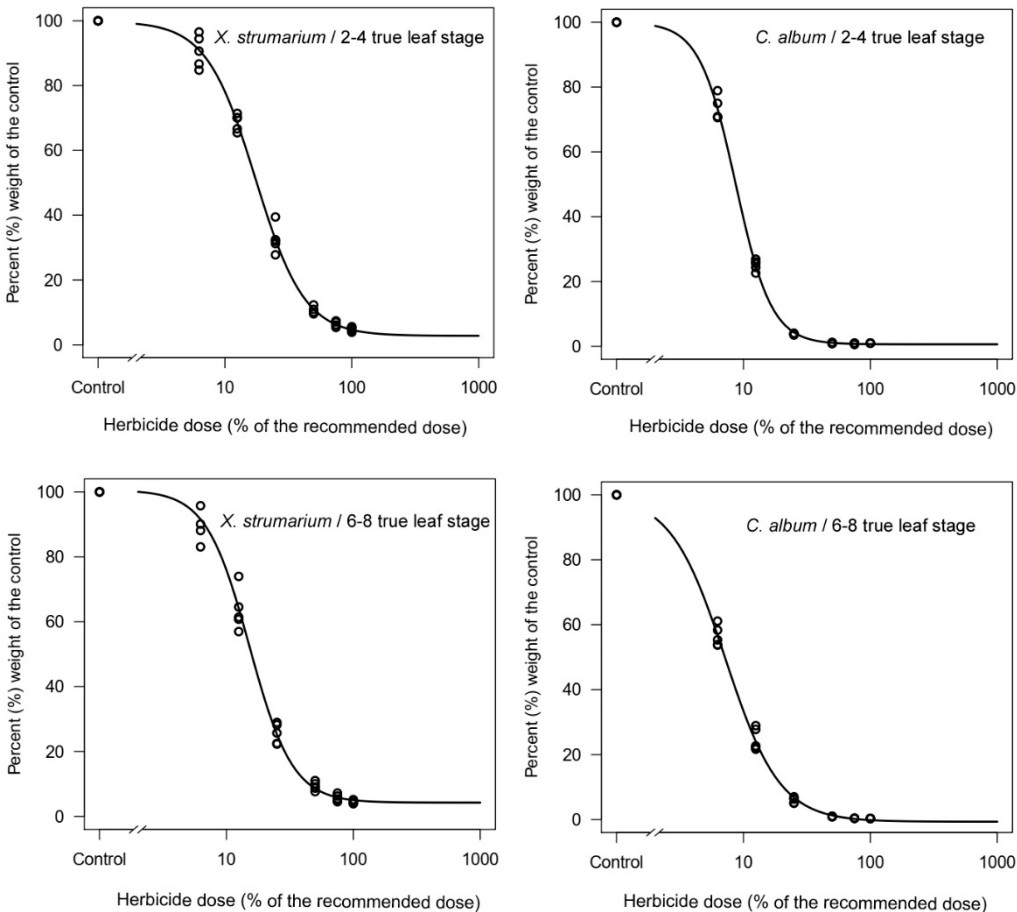

**Figure 5.** Dose–response curves of imazamox herbicide in controlling *X. strumarium* and *C. album* weeds at 2–4 and 6–8 true leaf stages.

The overall results for the plant height, leaf number and dry weight are in accordance with each other, as well as with some previous studies where the reduced doses of herbicides were noted to be successful for controlling weeds under different conditions [8,21,23,38–40]. However, the current results differ from several other studies in which higher doses were suggested for later leaf stages of the weeds [20,22]. The main reason for this difference is because of the characteristic of the imazamox herbicide. It is well known that the retention characteristic of a herbicide and the rate that is retained by the plants are highly important factors for the effectiveness of a herbicide [41]. The leaf area, leaf structure and spray-droplet contact angle also impact the effectiveness of foliar-applied herbicides [42]. In this case, the higher leaf area of the tested weeds at the 6–8 true leaf stage rather than the 2–4 true leaf stage could be the reason for the greater success at these leaf stages. Moreover, the recommended time of imazamox application was noted as being the 4–8 true leaf stage by the manufacturer [32]. Other than retention, the imazamox also had to be absorbed by the weeds and translocated up to its target site [43]. The systemic characteristic of the imazamox enables it to be quickly absorbed and translocated in the weeds. Thus, the leaf area and the temperature significantly increase the absorption and translocation of imazamox [44]. Furthermore, imazamox is a low-volatile compound [45] that can easily spread across the applied surface, and so the increase in leaf area can increase its absorption and efficiency. In a very recent study, Trezzi et al. [46] reported that the retention rate significantly impacts the efficiency of imazamox, which supports our results. However, it should be kept in mind that the effectiveness of herbicides is significantly affected by the weed development stage and density; if the stages increase further, the weeds become more tolerant and the efficacy of the herbicides decreases [47]. Therefore, the application time is crucial for reaching satisfactory control with the lowest herbicide doses. On the

other hand, it is well-known that besides the weed species and the growing period of the weeds, ecological conditions and weed populations [48] significantly affect the impact of herbicides. There can be important variation among the weed populations. Based on this information, it might be necessary to carry out similar studies with different weed populations to assess the sensitivity to imazamox at species level for both of the test weeds.

## 4. Conclusions

In conclusion, it was observed that the reduced doses of imazamox could be used for controlling *X. strumarium* and *C. album*. The results also suggested that the impact of imazamox increases with the increase in leaf area because of the higher rate of retention. It was found that 46.18 g ai/da of imazamox provided 90% success in the control of common cocklebur (*X. strumarium*) when it was applied at the 2–4 true leaf stage; a lower dose (36.11 g ai/da) was required for obtaining the same control when imazamox was applied at the 6–8 true leaf stage. Similar results were obtained for fat hen (*C. album*), but this weed was found to be more sensitive in comparison to *X. strumarium*. A 17.69 g ai/da imazamox dose was found to be adequate for reaching 90% control of *C. album* when it was applied at the 2-4 true leaf stage, and a 21.21 g ai/da imazamox dose was enough for controlling *C. album* at the 6–8 true leaf stage. The overall results suggested that in controlling weeds, both the weed species and weed development stages are important for the correct determination of the minimum herbicide dose, which could help to reduce the damage to the environment. The main reasons for this significant difference between the recommended dose and the effective doses ($ED_{90}$) are companies targeting more than one weed; considering different environmental conditions; and accounting for different weed development stages during the determination of the recommended doses of herbicides. For this reason, even if it seems costly, considering future generations and agricultural sustainability is extremely important to determine the recommended doses of herbicides for each individual weed species, and individual climatic conditions and growth periods.

**Author Contributions:** Conceptualization, R.G. and Ö.Y.; methodology, R.G. and Ö.Y.; formal analysis, R.G. and Ö.Y.; investigation, Ö.Y.; data curation, R.G. and Ö.Y.; writing—original draft preparation, Ö.Y.; writing—review and editing, R.G.; visualization, R.G.; supervision, R.G.; funding acquisition, R.G. All authors have read and agreed to the published version of the manuscript.

**Funding:** This research received no external funding.

**Data Availability Statement:** The data that support the findings of this study are available on request from the corresponding author.

**Acknowledgments:** The authors would like to thank Ibrahim Kahramanoğlu for helping with the data analysis.

**Conflicts of Interest:** The authors declare no conflict of interest.

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
