# Peer review of "Determination of Minimum Doses of Imazamox for Controlling Xanthium strumarium L. and Chenopodium album L. in Bean (Phaseolus vulgaris L.)"

_agronomy, doi:10.3390/agronomy12071557_

Round 1

Reviewer 1 Report

Corrections are done. Check if all is correct from editorial side.

Author Response

We are very thankful for the kind effort of the Reviewer for improving our paper. English language and style are checked. We hope that the final version would be acceptable for publication. 

Reviewer 2 Report

In the present manuscript, the authors aimed to determinate the minimum doses of imazamox required for the control of Xanthium strumarium and Chenopodium album, two of the most problematic weeds in bean fields. The experiments were carried out at two leaf stages (2-4 true leaf stage and 6-8 true leaf stage) and plants were treated with different imazamox doses ranging from 100% of the recommended dose to 25% of the recommended dose. Effects on the plant height and leaf number were monitored and a four-parameter log-logistic model was carried out to determine the minimum imazamox dose for the effective control of both weeds (90% reduction). The results showed that reduced doses of imazamox could be used for the control of both weeds and stressed the importance of considering the developmental stage when applying herbicides.

In general, the research design is well performed and conclusions are well supported by the results. However, I have some suggestions for the presentation of the data.

When presenting the results of the effect of imazamox on the plant height and number of leaves, having the results divided into three experiments makes it a bit confusing. I would propose to simplify the presentation of the data by, as it has been done for the log-logistic model, combining the results of the three experiments and presenting the resulting average.

Regarding the statistics of table 1, 2, 3 and 4, if I understood well, at each day, the mean of each dose has been compared. Please, specify which comparisons have been made, in the Material and Methods section. As the height and number of leaves of the plants at the beginning of the experiments is not the same, it cannot be compared the mean values at the different days, because the starting point is not the same for the different doses. I would suggest to perform the post hoc comparisons by comparing the mean values of each dose separately.

Some other comments:

-          - Page 1, line 18: “…herbicide symptoms on the weeds were recorded…”. No results regarding herbicide symptoms are presented, I would delete it from the abstract or refer to the symptoms in the results section.

-          - Page 1, line 24: “Result suggested that the increased are in leaf significantly reduce the imazamox requirement at the X. strumarium”. Change by: “Results suggested that the increase in leaf area reduces imazamox requirement for the control of X. strumarium”.

-          - Page 1, line 32: Change: “As of“ by “As for”

-          - Page 1, line 33: Change: “was” by is

-         - Page 1, line 37: “...of green production is the weeds,…”. Change by: “...of green production are weeds,…”

-          - Page 2, line 51: characteristics

-        - Page 2, line 52: “critical period and required minimum doses.” Change by “critical period and the required minimum doses of herbicides.”

-          - Page 2, lines 66-71: please, provide references

-          - Page 2, line 73: Imazamox

-          - Page 2, line 85: delete of

-          - Page 2, line 89: In the present study

-          - Page 3, line 100: 1.6 cm/cc. “cm/cc” is not a unit of volume, please correct it

-          - Page 3, line 103: please, specify which is the recommended dose of the herbicide

-          - Page 3, line 104: third experiment

-          - Page 3, line 110: needs of plants

-          - Page 3, line 111: change hectare by decare

-          - Page 3, line 124: It is not clear to me where are the pots placed. “Into field”, which field? Please explain it a bit.

-         - Page 3, line 126: “1st day” is the day of herbicide application or is it one day after imazamox application. In general, the day of treatment application is considered as the day 0, that is why I’m confused about it.

-          - Page 4, line 151: first experiment / second experiment

-          - Page 4, line 159: delete was

-          - Page 4, line 167: plant height

-          - Page 7, line 219: impact of imazamox

-          - Page 7, line 221: there is no decrease on plant height in experiment 2 (it is maintained in 8cm)

-          - Page 7, line 224: imazamox

-          - Page 8, line 232: in the current study

-          - Page 8, line 246: true leaf stage

-          - Page 9, line 259: among the treatments were first observed

-          - Page 9, line 264: when the imazamox was applied

-          - Page 10, line 276: the three highest doses

-          - Page 13, line 301: similar to

-          - Page 13, line 309: was applied

-          - Page 13, line 310: was applied

-          - Page 13, line 315: with respect to

-          - Page 13, line 318: 6-8 true leaf stage

-          - Page 13, line 319: 6-8 true leaf stage

-          - Page 15, line 354: 46.18 g ai/ha

-          - Page 16, line 373: of the tested weeds

-          - Page 16, line 382: impacts

-          - Page 15, line 386: decreases

-          - Page 17, line 402: 6-8 true leaf stage

Author Response

Thanks to the Reviewer for the valuable comments and corrections on the English language too. They were very helpful for correcting the paper and improving the presentation. We have carefully studied all comments of the Reviewer. A revised version of the paper with track changes was prepared and attached.

For the specific comments, we have combined the three experiments and used the resulting averages in the figures. For tables, we kept the data of each experiment to make it better for the readers to understand the text and be able to compare the experiments and results. For the data presented in the tables, we have performed Tukey’s HSD test at P<0.05 after ANOVA to compare the mean values of herbicide doses, at each measurement point. We did not compare the mean values of each dose separately. For this comparison, we have used the figures and log-logistic model, which made it possible to understand the changes in the weeds height and leaf number during the studies.

This manuscript is a resubmission of an earlier submission. The following is a list of the peer review reports and author responses from that submission.

Round 1

Reviewer 1 Report

In this manuscript, the authors aimed to determinate the minimum doses of imazamox for the control of two important weeds, Chenopodium album and Xanthium strumarium. The experiments were carried out at two leaf stages (2-4 true leaves and 6-8 true leaves) and plants were treated with different imazamox doses (100% of the recommended dose (RD), 75% RD, 50% RD, 25% RD and 0% RD). Effects on the plant height, leaf number and dry weight were monitored and a four-parameter log-logistic model was carried out to determine the minimum imazamox dose for the effective control of both weeds. The results showed that reduced doses could be used for controlling C. album and X. strumarium; and that the 6-8 true leaf stage is more sensitive to the herbicide than the 2-4 true leaf stage.

Overall, the experimental design is appropriate to test the hypothesis and interesting preliminary results have been obtained which indicate that the reduced imazamox doses are successful for controlling weeds. However, additional physiological parameters (photosynthetic parameters (SPAD, chlorophyll content…), ALS in vivo measurements, etc) should be included to support the results and make the study more robust. I am particularly concerned about the dose-response study, since a wider range of doses is required (minimum of 6-8 doses) in order to have solid results. I would suggest to include doses below the 25% of the RD. Related to this, I think that although the legend of the x-axes states that the herbicide doses are presented in “g ai / da”, data represent the herbicide doses in %, please revise it.

As for the comparison of the imazamox doses required for the control of the weeds depending on the developmental stage, interpretation of the data should be revised. In order to compare the effect of the herbicide on the 2-4 and 6-8 true leaf stages, as the values for plant height or leaf number at day 0 are quite different among both stages, it would be better to present the data  in “% with respect to the untreated plants” for its stage. When doing this, it is not that clear that imazamox has bigger effect on the 6-8 true leaf stage comparing to the 2-4 true leaf stage.

Finally, English must be revised to improve the readability of the text.

Considering all this, although I think that interesting preliminary data are presented, results are not robust enough to be published on its own. Thus, I recommend the rejection of the manuscript, additional experiments are needed and reinterpretation of the data should be done.

Author Response

Response to Reviewer 1 Comments

Thank you for your letter and for the reviewers’ comments on our manuscript entitled “Determination of Minimum Doses of Imazamox for Controlling Xanthium strumarium L. and Chenopodium album L. in Bean Fields” (ID: agronomy-1633054). All of these comments were very helpful for revising and improving our paper. We have studied these comments carefully and have made corresponding corrections that we hope will meet with your approval. A marked-up copy of our manuscript that highlights changes made to the final version with tracked changes is prepared and attached. The responses to the reviewers’ comments are provided below.

  1. R. Review Report (Reviewer 1)

In this manuscript, the authors aimed to determinate the minimum doses of imazamox for the control of two important weeds, Chenopodium album and Xanthium strumarium. The experiments were carried out at two leaf stages (2-4 true leaves and 6-8 true leaves) and plants were treated with different imazamox doses (100% of the recommended dose (RD), 75% RD, 50% RD, 25% RD and 0% RD). Effects on the plant height, leaf number and dry weight were monitored and a four-parameter log-logistic model was carried out to determine the minimum imazamox dose for the effective control of both weeds. The results showed that reduced doses could be used for controlling C. album and X. strumarium; and that the 6-8 true leaf stage is more sensitive to the herbicide than the 2-4 true leaf stage.

Overall, the experimental design is appropriate to test the hypothesis and interesting preliminary results have been obtained which indicate that the reduced imazamox doses are successful for controlling weeds. However, additional physiological parameters (photosynthetic parameters (SPAD, chlorophyll content…), ALS in vivo measurements, etc) should be included to support the results and make the study more robust. I am particularly concerned about the dose-response study, since a wider range of doses is required (minimum of 6-8 doses) in order to have solid results. I would suggest to include doses below the 25% of the RD. Related to this, I think that although the legend of the x-axes states that the herbicide doses are presented in “g ai / da”, data represent the herbicide doses in %, please revise it.

Our response: Thanks to the reviewer for the comments about our paper. As suggested by the reviewer different doses (lower or higher) can be tested. However, for our current study, results suggested that there is no need for that. This is the main aim of the dose-response curves. These dose-response analysis makes it possible to estimate the results of the non-tested doses. Thus, we do not think that it is necessary at this point to go back and test the lower doses. We believe that it is not entirely correct in this study to assess the effect of herbicides on the complex of physiological parameters of weeds. Since physiological indicators are more related to the strategy of forming and predicting the stability of agricultural crops. In this experiment, the emphasis was on the economically justified and effective use of different doses of herbicides depending on the type of weeds.

As for the comparison of the imazamox doses required for the control of the weeds depending on the developmental stage, interpretation of the data should be revised. In order to compare the effect of the herbicide on the 2-4 and 6-8 true leaf stages, as the values for plant height or leaf number at day 0 are quite different among both stages, it would be better to present the data in “% with respect to the untreated plants” for its stage. When doing this, it is not that clear that imazamox has bigger effect on the 6-8 true leaf stage comparing to the 2-4 true leaf stage.

Our response: Thanks to the reviewer for the recommendation. We have studies this comment carefully and replaced the existing figures with “% change with respect to the initial leaf number/plant height”. As suggested by the reviewer, this made it easier to compare the impacts of herbicide doses and the untreated control weeds. The existing information in the figures is also given as Tables to improve the understanding of the readers.

Finally, English must be revised to improve the readability of the text.

Our response: English language of the paper was checked and corrected throughout the paper.

Considering all this, although I think that interesting preliminary data are presented, results are not robust enough to be published on its own. Thus, I recommend the rejection of the manuscript, additional experiments are needed and reinterpretation of the data should be done.

Our response: We have studied the comments of both reviewers and we believe that the revised version of the paper can be accepted for publication, since it provides new and valuable information about the control of two important weed species (Chenopodium album and Xanthium strumarium) with reduced doses of imazamox.

In this work, an attempt was made to scientifically substantiate the use of a set of measures, including agricultural practices and chemicals (a reasonable integral protection system) aimed at the destruction of weeds and preventing their spread. These researches are an important direction of the section "Agriculture"

Response 1: Please provide your response for Point 1. (in red)

Reviewer 2 Report

Comments and suggestions for Authors:

  1. Article tittle: Determination of Minimum Doses of Imazamox for Controlling Xanthium strumarium L. and Chenopodium album L. in Green Bean (Phaseolus vulgaris L.)
  2. In whole paper use imazamox not Imazamox (text, figures and tables)
  3. Line 13: experiments
  4. Line 19: X. strumarium
  5. Line 25: Without BBCH development stage, add herbicide and ED90
  6. Line 34: potassium make
  7. Line 42: products instead drugs
  8. Line 57: climates
  9. Line 71: solubility
  10. Line 73: imazethapyr
  11. Line 85: gr/l
  12. Line 93: what "was high" means? Put % of content of organic matter!
  13. Line 108: gave the name and symbol of fan type nozzle
  14. Line 154: 35-38 cm
  15. Line 160 and 213: metribuzin
  16. Line 160 : Amaranthus retroflexus
  17. Line 161: Sinapis arvensis
  18. At all figures print A, B, C or D
  19.  Line167, 179, 186, 219, 252: C. album - print in italic
  20.  Line 189: name of the authors instead numbers [20,22]
  21. Table 1: no bold
  22. Line 375: Two times Kaur, P? Is this ok?
  23. Line 396: Symposium
  24. In conclusion explean why is so big differences bettwen recomended rate of imazamox and that from presented paper. 

Author Response

Response to Reviewer 1 Comments

Thank you for your letter and for the reviewers’ comments on our manuscript entitled “Determination of Minimum Doses of Imazamox for Controlling Xanthium strumarium L. and Chenopodium album L. in Bean Fields” (ID: agronomy-1633054). All of these comments were very helpful for revising and improving our paper. We have studied these comments carefully and have made corresponding corrections that we hope will meet with your approval. A marked-up copy of our manuscript that highlights changes made to the final version with tracked changes is prepared and attached. The responses to the reviewers’ comments are provided below.

2.R Review Report (Reviewer 2)

  1. Article tittle: Determination of Minimum Doses of Imazamox for Controlling Xanthium strumarium  and Chenopodium album L. in Green Bean (Phaseolus vulgaris L.)

Our response: Thanks to the reviewer. We have revised the title as suggested by the reviewer.

  1. In whole paper use imazamox not Imazamox (text, figures and tables)

Our response: It was changed and corrected throughout the paper.

  1. Line 13: experiments

Our response: It was corrected.

  1. Line 19:  strumarium

Our response: It was corrected.

  1. Line 25: Without BBCH development stage, add herbicide and ED90

Our response: It was corrected.

  1. Line 34: potassium make

Our response: It was corrected.

  1. Line 42: products instead drugs

Our response: It was corrected.

  1. Line 57: climates

Our response: It was corrected.

  1. Line 71: solubility

Our response: It was corrected.

  1. Line 73: imazethapyr

Our response: It was corrected.

  1. Line 85: gr/l

Our response: It was corrected.

  1. Line 93: what "was high" means? Put % of content of organic matter!

Our response: This information was added to the text.

  1. Line 108: gave the name and symbol of fan type nozzle

Our response: The name and symbol of fan type nozzle are given in the text.

  1. Line 154: 35-38 cm

Our response: It was corrected.

  1. Line 160 and 213: metribuzin

Our response: They were corrected.

  1. Line 160 : Amaranthus retroflexus

Our response: It was corrected.

  1. Line 161: Sinapis arvensis

Our response: It was corrected.

  1. At all figures print A, B, C or D

Our response: The letters were given on the figures. The figures were all changed with the “% change with respect to the initial leaf number/plant height” as suggested by reviewer 2.

  1. Line167, 179, 186, 219, 252: C. album - print in italic

Our response: They were all corrected in typing.

  1. Line 189: name of the authors instead numbers [20,22]

Our response: The surnames of the authors are given as suggested by the reviewer.

  1. Table 1: no bold

Our response: It was corrected.

  1. Line 375: Two times Kaur, P? Is this ok?

Our response: We have checked the author names, they are correct: Paawan Kaur, Pervinder Kaur and Navjyot Kaur and et al.

  1. Line 396: Symposium

Our response: It was added.

  1. In conclusion explain why is so big differences between recommended rate of imazamox and that from presented paper. 

Our response: Thanks to the reviewer. We have explained this difference in the conclusions section.

Response 2: Please provide your response for Point 2. (in red)

Round 2

Reviewer 1 Report

Although authors have addressed some of my comments, they did not solved what worries me the most. I still have serious concerns regarding the dose-response study. The experiment has serious flaws. On the one hand, fitting the log-logistic curve requires five to seven doses, and in this experiment, only four doses have been used. To get a robust dose-response curve, the applied doses should cover the whole range of responses from no visible effects to complete kill of plants and they should include doses near the I50. I believe that the doses used in this experiment are very aggressive and that they do not accurately describe data, especially at the low extreme of doses. In the present study, plants are severely affected by all the applied doses, and it is not possible to get a good dose-response curve (with the typical sigmoidal shape). This is particularly evident in the case of C. album, where figures show that the dry weight is almost the same in the four treatments (near 0, which means that all plants are death), indicating that the applied doses have not been properly selected to obtain a good dose-response curve. I believe that 28 days is too late to measure the gradual effect of the different herbicide doses on the dry weight. I would suggest to repeat the experiment including one or two more doses, and I would recommend to harvest the plants earlier.

In addition, I still think that in Figure 5, the herbicide doses are represented in percentage and not in “g of a.i per da”, as it is indicated in the legend. Please, check it.

Some other comments:

  • Please, use the International System of Units, thus, use ha instead of da
  • Page 2 lines 64-68. Imazamox does not inhibit ALS production, instead it inhibits ALS activity and thus, blocks the production of branched-chain amino acids.
  • There are still some English spelling mistakes to be corrected

Author Response

Although authors have addressed some of my comments, they did not solved what worries me the most. I still have serious concerns regarding the dose-response study. The experiment has serious flaws. On the one hand, fitting the log-logistic curve requires five to seven doses, and in this experiment, only four doses have been used. To get a robust dose-response curve, the applied doses should cover the whole range of responses from no visible effects to complete kill of plants and they should include doses near the I50. I believe that the doses used in this experiment are very aggressive and that they do not accurately describe data, especially at the low extreme of doses. In the present study, plants are severely affected by all the applied doses, and it is not possible to get a good dose-response curve (with the typical sigmoidal shape). This is particularly evident in the case of C. album, where figures show that the dry weight is almost the same in the four treatments (near 0, which means that all plants are death), indicating that the applied doses have not been properly selected to obtain a good dose-response curve. I believe that 28 days is too late to measure the gradual effect of the different herbicide doses on the dry weight. I would suggest to repeat the experiment including one or two more doses, and I would recommend to harvest the plants earlier.

Thanks to the reviewer for his/her recommendations and concerns. We have tried to explain in our first revision, but we could not. First of all, there were 5 doses in our experiment, not 4. The zero dose, as a control, also provides data for dose-response analysis. The shape of the dose-response curve is greatly influenced by the distribution of raw data it describes. And yes, the appropriate selection of herbicide doses is critical for dose-response analysis (Streibig et al. 1993). For example, a sufficiently high dose stops growth or might kill the plants, resulting in data grouping at the lower end of the curve, but sufficiently low doses have no effect on the plant resulting in data grouping around the upper end of the curve. The line between the two ends of the curve is usually straight (Knezevic et al. 1998).

According to Knezevic et al. (2007), a four-parameter log-logistic curve (which was used in our experiments) could be described by four doses; one dose for each parameter. This is what we have done in our experiments. There are several studies, which support our experimental methodologies. Following references used 4 doses (25%, 50%, 75% and 100%) and a control dose (0%) in their experimental studies:

  • Diyanat, M., & Ghasemkhan-Ghajar, F. IIntegrated weed control in corn (Zea mays L.) through combinations of seed priming and reduced dosages of various commonly used herbicides. International Journal of Advanced Biological and Biomedical Research8(3), 291-302.
  • Elahinejad, M., Oveisi, M., & Rahimian, H. (2021). Effect of planting density and dose of Imaztapir herbicide on weed control in mixed cultivation of standing and creeping cultivars of red bean (Phaseolus vulgaris). Iranian Journal of Weed Science17(1).
  • Khaliq, A., Matloob, A., Tanveer, A., Areeb, A., Aslam, F., & Abbas, N. (2011). Reduced doses of a sulfonylurea herbicide for weed management in wheat fields of Punjab, Pakistan. Chilean journal of agricultural research71(3), 424.
  • Naby, K. Y., & Ali, K. A. (2020). a. Integrated weed management in wheat crops by applying sorghum aqueous extract and reduced herbicide dose. Plant Archives20(2), 3618-3623.
  • Oveisi, M., Kaleibar, B. P., Mashhadi, H. R., Müller-Schärer, H., Bagheri, A., Amani, M., ... & Masoumi, D. (2021). Bean cultivar mixture allows reduced herbicide dose while maintaining high yield: A step towards more eco-friendly weed management. European Journal of Agronomy122, 126173.
  • Zhang, J., Zheng, L., Jäck, O., Yan, D., Zhang, Z., Gerhards, R., & Ni, H. (2013). Efficacy of four post-emergence herbicides applied at reduced doses on weeds in summer maize (Zea mays L.) fields in North China Plain. Crop Protection, 52, 26-32.

Moreover, there are some other studies, where only 3 doses were tested:

  • Barros, J. F., Basch, G., & de Carvalho, M. (2007). Effect of reduced doses of a post-emergence herbicide to control grass and broad-leaved weeds in no-till wheat under Mediterranean conditions. Crop Protection26(10), 1538-1545.
  • Piekarczyk, M. (2021). Response of diversified doses of herbicide mixture on weed control and yield in white lupine. Acta Scientiarum Polonorum Agricultura20(1), 17-23.

Of course, there are some studies, which sued 5 doses + control in their studies (i.e. KahramanoÄŸlu and Uygur, 2010). Therefore, the important thing here is not the number of doses used; however, is the suitability of the obtained data with the tested model. The P-values of the lack-of-fit test are used to test the acceptability of the data with the model (Knezevic et al., 2007; KahramanoÄŸlu and Uygur, 2010; Ritz et al., 2015). It is clear from our results (where the P-values are all not significant, P > 0.05) (see Table 5), that all our data provided an acceptable fit to the dose-response curve. So, our data best fits to the model and there is no need to include lower doses into our experimental design.

The timing of data collection depends on the hypothesis and objectives of the experiment (Knezevic et al. 2002). It is generally recommended for the 4 weeks period for the minimum dose studies (Knezevic et al. 2002; KahramanoÄŸlu and Uygur, 2010)

References:

KahramanoÄŸlu, İ., & Uygur, F. N. (2010). The effects of reduced doses and application timing of metribuzin on redroot pigweed (Amaranthus retroflexus L.) and wild mustard (Sinapis arvensis L.). Turkish Journal of Agriculture and Forestry34(6), 467-474.

Knezevic, S. Z., Streibig, J. C., & Ritz, C. (2007). Utilizing R software package for dose-response studies: the concept and data analysis. Weed Technology, 21(3), 840-848.

Knezevic, Z. S., S. P. Evans, E. Blankenship, R. VanAcker, and J. L. Lindquist. 2002. Critical period of weed control: the concept and data analysis. Weed Sci. 50:773-786

Knezevic, Z. S., P. H. Sikkema, F. Tardif, A. S. Hamill, K. Chandler, and C. J. Swanton. 1998. Biologically effective dose and selectivity of RPA 201772 (isoxaflutole) for preemergence weed control in corn. Weed Technol. 12:670-676

Ritz, C., Baty, F., Streibig, J. C., & Gerhard, D. (2015). Dose-response analysis using R. PloS one, 10(12), e0146021.

Streibig JC, Rudermo M, Jensen JE (1993) Dose-response curves and statistical models. In: JC Streibig & P Kudsk (Eds.), Herbicide Bioassays. CRC Press, Boca Raton, USA, pp. 30-55.

In addition, I still think that in Figure 5, the herbicide doses are represented in percentage and not in “g of a.i per da”, as it is indicated in the legend. Please, check it.

Thanks to the Reviewer for the comment. We have corrected and changed labels into percentage.

Some other comments:

  • Please, use the International System of Units, thus, use ha instead of da

We have changed the units from da to ha.

  • Page 2 lines 64-68. Imazamox does not inhibit ALS production, instead it inhibits ALS activity and thus, blocks the production of branched-chain amino acids.

Thanks to the reviewer for the comment. We have corrected the sentence.

  • There are still some English spelling mistakes to be corrected

We have made some additions corrections within the revised version of the manuscript.
